# Long-term neurodevelopmental outcome in children born after vacuum-assisted delivery compared with second-stage caesarean delivery and spontaneous vaginal delivery: a cohort study

Stefhanie Romero ,[1,2] Katarina Lindström,[3,4] Johanna Listermar,[5] Magnus Westgren,[1] Gunilla Ajne[1,2]

For numbered affiliations see end of article.

**Correspondence to**
Dr Stefhanie Romero; stefhanie.romero@ki.se

## ABSTRACT

**Objective** To evaluate long-term neurodevelopment in children born after low-or mid-station vacuum-assisted delivery (VAD) compared with children delivered by second-stage caesarean delivery (SSCD) or spontaneous vaginal delivery (SVD).

**Design** Cross-sectional cohort study.

**Setting** Two delivery wards, Karolinska University Hospital, Sweden.

**Patients** 253 children born by low-station or mid-station VAD, 247 children born after an SVD, and 86 children born via an SSCD accepted to participate.

**Interventions** The Five-to-Fifteen questionnaire was used as a validated screening method for neurodevelopmental difficulties, assessed by parents.

**Main outcomes measures** Results in the Five-to-Fifteen questionnaire. In addition, registered neurodevelopmental ICD-10 diagnoses were collected. Regression analyses estimated associations between delivery modes.

**Results** Children born after VAD exhibited an increased rate of long-term neurodevelopmental difficulties in motor skills (OR 2.2, 95% CI 1.3 to 3.8) and perception (OR 1.7, 95% CI 1.002 to 2.9) compared with SVD. Similar findings were seen in the group delivered with an SSCD compared with SVD (motor skills: OR 3.3, 95% CI 1.8 to 6.4 and perception: OR 2.3, 95% CI 1.2 to 4.4). The increased odds for motor skills difficulties after VAD and SSCD remained after adjusting for proposed confounding variables. There were significantly more children in the VAD group with registered neurodevelopmental ICD-10 diagnoses such as attention deficit/hyperactivity disorders.

**Conclusions** The differences in long-term neurodevelopmental difficulties in children delivered with a VAD or SSCD compared with SVD in this study indicate the need for increased knowledge in the field to optimise the management of second stage of labour.

## INTRODUCTION

The vacuum extractor is used in cases of dystocia or fetal distress during the second stage of labour.[1] The incidence of vacuum-assisted delivery (VAD) varies worldwide,

---

### WHAT IS ALREADY KNOWN ON THIS TOPIC

⇒ There is an evident lack of information on the long-term neurodevelopmental difficulties in children born after a vacuum-assisted delivery.

### WHAT THIS STUDY ADDS

⇒ Children born after low-station or mid-station vacuum-assisted deliveries and second-stage caesarean deliveries may be at an increased odds of long-term neurodevelopmental difficulties in comparison to those born after a spontaneous vaginal delivery.

⇒ Children delivered with a vacuum-assisted delivery may have more attention deficit/hyperactivity disorders than the other two delivery groups.

### HOW THIS STUDY MIGHT AFFECT RESEARCH, PRACTICE OR POLICY

⇒ These striking results need to be promptly investigated, preferably in prospective studies, with a revision of the clinical guidelines.

---

representing up to 3% among all vaginal births in the USA,[2] and 5%–10% in Europe, Canada and Australia/New Zealand.[3–7]

Serious fetal complications are rare after VAD, but appears to have a higher rate of intracranial haemorrhage (0.1%–0.8%)[8] and subgaleal hematomas (0.4%–2.5%)[9] compared with spontaneous vaginal delivery (SVD).[10 11] Results are not consistent when comparing VAD to other instrumental delivery modes, including second-stage caesarean delivery (SSCD).[9 12 13] Some studies indicate an increased risk of birth trauma in the VAD group,[14] while others report no differences in severe neonatal outcomes.[15 16] The most appropriate comparison to VAD is SSCD since both are optional delivery methods to an SVD at this point.

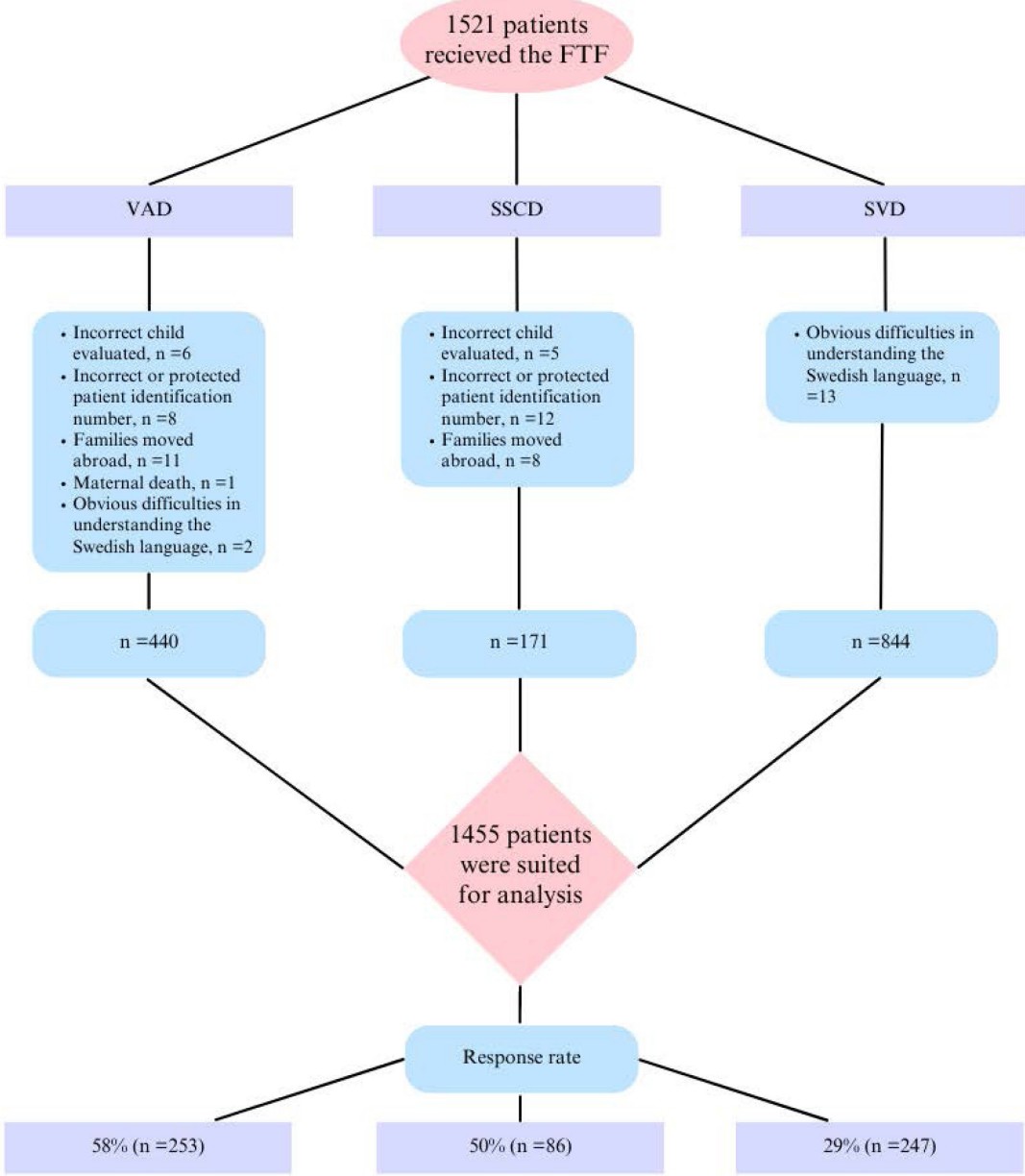

**Figure 1** Study flow chart. FTF, Five-to-Fifteen questionnaire; SSCD, second-stage caesarean delivery; SVD, spontaneous vaginal delivery; VAD, vacuum-assisted delivery.

VAD is classified according to fetal head station in the birth canal into mid, low and outlet stations, the last of which is considered a low-risk VAD.[17 18] Previous research has not always taken this stratification into account, which could be one reasons behind the divergent results.

Considering the large number of children exposed to VAD, there is limited information regarding long-term outcomes. Existing studies have not revealed any significant differences in neurodevelopment outcome.[19–25]

The aim of this study was to study neurodevelopment in children delivered at term by low-station or mid-station VAD compared with children born after SSCD and SVD.

## METHODS
### Study design and study population

This was a cross-sectional cohort study, including all women delivered with a low-station or mid-station VAD at Karolinska University Hospital Huddinge, Sweden between 2007 and 2009 (n=493). Each individual extraction protocol was examined to confirm the classification of fetal station.[6 18 26] Exclusion criteria were outlet station VADs, preterm delivery <37+0 gestational weeks (n=19), body mass index <18 kg/m$^2$ and multiple pregnancies (n=6), with a final number of 468 children.

The SVD group consisted of women and their children delivered at Karolinska University Hospital Huddinge. This group was included 3 years after the VAD group but

**Table 1** Descriptive analysis according to delivery mode

| | VAD (n=253) | SVD (n=247) | SSCD (n=86) | P value |
|---|---|---|---|---|
| **Maternal characteristics** | | | | |
| Nulliparous* † | 208 (82%) | 151 (61%) | 67 (78%) | <0.001‡§ <0.01¶ § 0.38** |
| Previous caesarean* † | 23 (9%) | 4 (2%) | 11 (13%) | <0.001‡§ <0.001¶ § 0.32** |
| Post high school education* † | 166 (66%) | 185 (75%) | 61 (71%) | <0.05‡§ 0.5¶ 0.4** |
| BMI (kg/m$^2$)†† ‡‡ | 23 (18–50) | 23 (18–35) | 23 (18–37) | 0.42‡ <0.01¶ § <0.05** § |
| Gestational length, days†† ‡‡ | 281 (261–295) | 279 (259–294) | 281 (259–295) | <0.001‡§ <0.001¶ § <0.05** § |
| First stage of labour, hours§§ ¶¶ | 8.8 (±4.6) | 4.2 (±3.7) | 8.4 (±5.2) | <0.001‡§ <0.001¶ § 0.53** |
| Second stage of labour, hours†† ‡‡ | 2 (0–6.5) | 0.5 (0–5) | 5 (0–17) | <0.001‡§ <0.001¶ § <0.001** § |
| **Delivery characteristics** | | | | |
| Indication OFHR | 108 (43%) | | 22 (26%) | <0.01** § |
| Indication dystocia | 145 (57%) | | 60 (74%) | |
| Epidural* † | 183 (72%) | 78 (32%) | 69 (80%) | <0.001‡§ <0.001¶ § 0.2** |
| Oxytocin* † | 242 (96%) | 99 (40%) | 83 (97%) | <0.001‡§ <0.001¶ § 0.7** |
| Position OAP* † | 221 (90%) | 237 (96%) | 61 (71%) | <0.05‡§ <0.001¶ § <0.001** § |
| Shoulder dystocia* † | 11 (4%) | 0 | – | 0.001‡§ |
| **Perinatal characteristics** | | | | |
| Birth weight, g§§ ¶¶ | 3668 (±447) | 3578 (±429) | 3836 (±494) | <0.05‡§ <0.001¶ § <0.01** § |
| Gender male* † | 115 (46%) | 119 (48%) | 36 (42%) | 0.5‡ 0.3¶ 0.6** |
| NICU admission* † | 40 (16%) | 1 (0.4%) | 5 (6%) | <0.001‡§ 0.001¶ § <0.05** § |
| pH a. umbilicalis<7.00* † | 4 (2%) | 0 | 0 | <0.05‡§ 0.2** |
| Apgar<7 at 5 min* † | 9 (4%) | 0 | 0 | <0.01‡§ 0.08** |

*n (%).
†χ2 test.
‡VAD versus SVD.
§Statistically significant at p<0.05.
¶SSCD versus SVD.
**VAD versus SSCD.
††Median (min-max).
‡‡Wilcoxon rank sum test.
§§mean±SD.
¶¶Student's t-test.
BMI, body mass index; NICU, neonatal intensive care unit; OAP, occipital-anterior position; OFHR, ominous fetal heart rate; SSCD, second-stage caesarean delivery; SVD, spontaneous vaginal delivery; VAD, vacuum-assisted delivery.

**Table 2** Age distribution among the children for each delivery mode

| Age (years) | VAD (n=253) | SVD (n=247) | SSCD (n=86) |
|---|---|---|---|
| 6 | – | – | 11 (13%) |
| 7 | 50 (20%) | 62 (25%) | 13 (15%) |
| 8 | 82 (32%) | 88 (36%) | 16 (19%) |
| 9 | 104 (41%) | 81 (33%) | 8 (9%) |
| 10 | 17 (7%) | 16 (6%) | 12 (14%) |
| 11 | – | – | 8 (9%) |
| 12 | – | – | 17 (20%) |
| 13 | – | – | 1 (1%) |
| Mean age at FTF completion | 8 (±0.9) | 8 (±0.9) | 9 (±2.2) |

N (%).
FTF, Five-to-Fifteen questionnaire; SSCD, second-stage caesarean delivery; SVD, spontaneous vaginal delivery; VAD, vacuum-assisted delivery.

matched by calendar date and time of birth (2010–2012, n=870). After applying the exclusion criteria, the final group consisted of 869 children. In addition, we identified all cases of SSCD with fetal heads at least at station 0 by scrutinising the medical records. As this delivery mode was expected to be a rarer event, we included all women at both delivery wards at the Karolinska University Hospital, Huddinge and Solna, between 2007 and 2014 (n=184).

The VAD followed national and international guidelines consisting of limiting the procedure to 20 min, a maximum of two cup detachments and six pulls.[6 18 26 27] The use of a Bird metal cup (50 mm) and a vacuum pressure of 80 kPa was standard. Both the VAD and the SSCD procedures were exclusively carried out by physicians. SVDs were managed by a midwife, according to routine procedures in the Swedish Health Care system.

A predefined response analysis was included in the study protocol for the VAD group.

### Long-term follow-up

The Five-to-Fifteen questionnaire (FTF) was used as a primary long-term follow-up. It is a validated screening method for behavioural and neurodevelopmental difficulties.[28] It is aimed at parents, covering different aspects of the child's functioning. It is subdivided into three age groups (6–8 years, 9–12 years and 13–15 years) and presented for boys and girls of a Swedish norm group.[28–31]

It comprises 181 items divided into 8 main domains (motor skills, executive functions, perception, memory, language, learning, social skills and emotional/behavioural problems). The 90th percentile is calculated and compared with normative data, used as a cut-off for obvious reported difficulties.

Study participants were contacted by mail including the FTF in Swedish. After 2 months, a reminder was sent out to the non-responding families, and later a final request.

A paediatric neurologist with no information on delivery mode reviewed the unidentified results by group (</≥90th percentile) for each domain to account for non-random misclassification. The results were transferred to 5–15.org for calculation.

### Data acquisition

Maternal, obstetric and neonatal characteristics were collected from computerised delivery and neonatal charts (Obstetrix).

Registered ICD-10 (International Classification of Diseases 10th Revision) diagnoses for the participating children were acquired from the national patient register at the Swedish National Board of Health and Welfare (SNBHW) until 2021. Included ICD-10 codes were F70–73, F78–79 (mental retardation), F80–84, F88.9 and F89.9 (disorders of the psychological development), F90–95, F98–99 (behavioural and emotional disorders), R48 (dyslexia and other symbolic dysfunctions), G80 (cerebral palsy), and G40 and G41 (episodic and paroxysmal disorders). The Swedish national patient register includes information on inpatient care since 1964 and outpatient specialist care since 2001. The validity of most diagnoses, including psychiatric disorders, has proven to be high.[32]

Maternal educational level was collected from the longitudinal integration database for health insurance and market studies, Statistics Sweden (Statistiska centralbyrån). The SUN2000 classification (Svensk UtbildningsNomenklatur) is divided into categories: (0) preschool education; (1) pre high school education <9 years; (2) pre high school education 9–10 years; (3) high school education; (4) post high school education <2 years; (5) post high school education ≥2 years and (6) doctoral studies/PhD. A subdivision of post high school studies yes/no was done (levels 4–6).

### Outcomes

The primary outcome was the FTF score for each main domain according to delivery mode. Secondary outcomes were the incidence of scoring ≥90th percentile in ≥2 main domains in the FTF for each delivery mode, and the incidence of an ICD-10 paediatric neurodevelopmental diagnosis.

### Statistical analysis

IBM SPSS Statistics (V.27) was used reporting 95% CIs and p values. Descriptive statistics are presented as mean±SD, median (min-max), and numbers and percentages. The $\chi^2$ test was used for dichotomous data. Independent samples Student's t-test or Wilcoxon rank sum tests were used for continuous variables. Univariate regression analyses were performed comparing: (1) VAD versus SVD; (2) SSCD versus SVD and (3) VAD versus SSCD, with the last

**Table 3** Number and percentages of children scoring ≥90th percentile in the FTF per main domain; a univariate and multivariable logistic regression analysis

| Main domain | VAD n/total (%) | SVD n/total (%) | SSCD n/total (%) | OR (95% CI) | aOR (95% CI) |
|---|---|---|---|---|---|
| Motor skills | 47/253 (19%) | 23/246 (9%) | 22/86 (26%) | 2.2 (1.3 to 3.8)* | 2.2 (1.1 to 4,7)* |
| | | | | 3.3 (1.8 to 6.4)† | 3.2 (1.3 to 7.5)† |
| | | | | 0.7 (0.4 to 1.2)‡ | 0.6 (0.3 to 1.1)‡ |
| Executive functions | 40/253 (16%) | 34/247 (14%) | 10/86 (12%) | 1.2 (0.7 to 1.9)* | 1.5 (0.8 to 3.1)* |
| | | | | 0.8 (0.4 to 1.8)† | 1.2 (0.5 to 3.2)† |
| | | | | 0.8 (0.4 to 1.8)‡ | 1.2 (0.5 to 2.5)‡ |
| Perception | 42/253 (17%) | 26/247 (11%) | 18/86 (21%) | 1.7 (1.002 to 2.9)* | 1.9 (0.9 to 4.1)* |
| | | | | 2.3 (1.2 to 4.4)† | 1.9 (0.8 to 4.7)† |
| | | | | 0.8 (0.4 to 1.4)‡ | 0.6 (0.3 to 1.2)‡ |
| Memory | 29/253 (11%) | 26/247 (11%) | 5/86 (6%) | 1.1 (0.6 to 1.9)* | 1.1 (0.5 to 2.4)* |
| | | | | 0.5 (0.2 to 1.4)† | 0.5 (0.1 to 1.6)† |
| | | | | 2.1 (0.8 to 5.6)‡ | 1.9 (0.7 to 5.1)‡ |
| Language | 31/253 (12%) | 22/247 (9%) | 6/86 (7%) | 1.4 (0.8 to 2.5)* | 1.7 (0.7 to 3.8)* |
| | | | | 0.8 (0.3 to 2)† | 0.6 (0.2 to 1.8)† |
| | | | | 1.9 (0.8 to 4.6)‡ | 1.7 (0.7 to 4.2)‡ |
| Learning | 30/195 (15%) | 23/192 (12%) | 7/62 (11%) | 1.3 (0.8 to 2.4)* | 1.6 (0.7 to 3.5)* |
| | | | | 0.9 (0.4 to 2.3)† | 0.7 (0.2 to 2.2)† |
| | | | | 1.4 (0.6 to 3.4)‡ | 1.2 (0.7 to 3.1)‡ |
| Social skills | 30/252 (12%) | 19/246 (8%) | 6/81 (7%) | 1.6 (0.9 to 3)* | 1.2 (0.5 to 2.6)* |
| | | | | 0.9 (0.4 to 2.5)† | 0.6 (0.2 to 1.9)† |
| | | | | 1.7 (0.7 to 4.2)‡ | 1.6 (0.6 to 4.1)‡ |
| Emotional/behavioural problems | 45/253 (18%) | 29/247 (12%) | 13/86 (15%) | 1.6 (0.9 to 2.7)* | 1.4 (0.7 to 2.8)* |
| | | | | 1.3 (0.7 to 2.7)† | 1.4 (0.5 to 3.4)† |
| | | | | 1.2 (0.6 to 2.4)‡ | 1.1 (0.6 to 2.2)‡ |

The multivariable logistic regression analysis shows adjusted OR for parity, maternal educational level, gestational length, oxytocin usage in VAD versus SVD and SSCD versus SVD, and indication OFHR in VAD versus SSCD.
*VAD versus SVD.
†SSCD versus SVD.
‡VAD versus SSCD.
aOR, adjusted OR; FTF, Five-to-Fifteen questionnaire; OFHR, ominous fetal heart rate; SSCD, second-stage caesarean delivery; SVD, spontaneous vaginal delivery; VAD, vacuum-assisted delivery.

**Table 4** Number and percentages of children scoring ≥90th percentile in the FTF in ≥2 domains; a univariate logistic regression analysis

| | VAD n/total (%) | SVD n/total (%) | SSCD n/total (%) | OR (95% CI) |
|---|---|---|---|---|
| ≥2 domains | 65/87 (75%) | 37/73 (49%) | 22/31 (71%) | 1.96 (1.3 to 3.1)* |
| | | | | 1.95 (1.1 to 3.6)† |
| | | | | 1.00 (0.6 to 1.8)‡ |

*VAD versus SVD.
†SSCD versus SVD.
‡VAD versus SSCD.
FTF, Five-to-Fifteen questionnaire; SSCD, second-stage caesarean delivery; SVD, spontaneous vaginal delivery; VAD, vacuum-assisted delivery.

**Table 5**  Children's diagnoses according to ICD-10

|  | VAD (n=253) | SVD (n=247) | SSCD (n=86) |
|---|---|---|---|
| Communication disorders (F80) | 4 (2%) | 1 (0.4%) | 0 |
| Specific developmental disorder of motor function (F829) | 2 (0.8%) | 0 | 0 |
| Autism spectrum disorders (F84) | 12 (5%) | 6 (2%) | 0 |
| Unspecified disorder of psychological development (F899) | 0 | 0 | 1 (1%) |
| Attention deficit hyperactivity disorder (F90) | 23 (9%) | 8 (3%) | 1 (1%) |
| Oppositional defiant disorder (F913) | 1 (0.4%) | 0 | 0 |
| Tic disorders (F95) | 2 (1%) | 0 | 0 |
| Enuresis (F980) | 4 (2%) | 4 (2%) | 1 (1%) |
| Encopresis (F981) | 1 (0.4%) | 0 | 3 (3%) |
| Developmental and emotional disorders (F98) | 2 (0.8%) | 0 | 1 (1%) |
| Unspecified mental disorder (F999) | 3 (1%) | 2 (0.8%) | 0 |
| Epilepsy (G40) | 1 (0.4%) | 2 (0.8%) | 1 (1%) |
| Neurodevelopmental diagnosis | 36 (14%) | 15 (6%) | 7 (8%) |
| Child with ≥2 neurodevelopmental diagnoses | 20 (8%) | 6 (2%) | 2 (2%) |

N (%).
ICD-10, International Classification of Diseases 10th Revision; SSCD, second-stage caesarean delivery; SVD, spontaneous vaginal delivery; VAD, vacuum-assisted delivery.

one serving as the reference group for each main domain in the FTF scores.

To assist in choosing the variables in the multivariable logistic regression analyses, a directed acyclic graph was designed, investigating potential confounders in the relationship between the exposure (delivery mode) and outcome (response in the FTF) (online supplemental figure S1). The variables finally included in the analyses were parity, maternal educational level, gestational length (in days), the indication ominous fetal heart rate for the comparison VAD versus SSCD and the use of oxytocin in the comparison VAD versus SVD and SSCD versus SVD.

### Public and patient involvement statement
There was no public or patient involvement. Written consent was signed on the front page of the FTF questionnaire.

### RESULTS
The annual delivery rate during the study period was 4500 at Karolinska University Hospital Huddinge and 4000 at Solna, including 10% emergency caesarean sections and 7%–9% VADs (50% at outlet station and 7.1% failed extractions). Both delivery wards are managed by the same organisation.

A total of 1521 families received the FTF questionnaire. After exclusion (figure 1), 1455 families were invited to participate (VAD n=440, SVD n=844 and SSCD n=171). The response rate was 58% in the VAD group (n=253), 29% in the SVD group (n=247) and 50% in the SSCD group (n=86).

The descriptive analysis on responders and non-responders in the VAD group (online supplemental table 1) showed no differences except more mid-station VADs in the non-response group (71% vs 59%).

A descriptive analysis according to delivery mode is shown in table 1.

Out of all VADs, 13% failed (n=33) and were delivered by subsequent delivery modes (forceps n=8, SSCD n=17 and forceps plus SSCD n=8). The failed VADs were included in the VAD group.

### The FTF questionnaire
A basic characteristics comparison between children with obvious difficulties in one or more domains according to the answers in the FTF (≥90th percentile) and children without difficulties (<90th percentile), according to normative data, was performed for each delivery mode. For all delivery modes, there were statistically significantly more women with a post -high school education in the group with children scoring 90th percentile. Apart from this, there were more subjectively heavy extractions among children scoring ≥90th percentile born after VAD (p<0.05). In the SSCD group, there were more males among children scoring <90th percentile (p<0.01). Table 2 presents the age distribution of the children delivered by each delivery mode. The FTF was assessed in 2017 in the VAD group, in 2020 in the SVD group, and in 2019–2020 in the SSCD group.

The univariate regression analyses are presented in table 3. There was a higher proportion of children scoring to have problems linked to motor skills and perception in the VAD group compared with the SVD group. The

same results were seen when comparing SSCD to SVD. There were no calculated differences between the VAD and SSCD groups. In the multivariable logistic regression analysis, the calculated aOR remained increased for problems with motor skills in children delivered with a VAD or SSCD compared with children delivered with an SVD.

There were significantly more children with obvious difficulties in two or more domains in the VAD and SSCD groups compared with the SVD group (table 4).

The incidence of the diagnoses acquired from the SNBHW is presented in table 5. There are more children with an attention deficit/hyperactivity disorder (ADHD) delivered with VAD compared with SVD (p<0.01) and to ECD (p<0.05). Furthermore, children born by VAD had a higher incidence of neurodevelopmental diagnoses than children born by SVD (p<0.001).

## DISCUSSION
### Main findings
According to the results from the studied cohort, low-station and mid-station VADs imply increased odds of long-term neurodevelopmental difficulties in motor skills and perception between 7 and 10 years of age compared with spontaneously delivered children. Similar findings were seen in the group delivered with an SSCD. After adjusting for the proposed confounding variables, the increased odds for difficulties in motor skills after a VAD or an SSCD was maintained. The incidence of ADHD was higher among children born after a VAD.

Considering the relatively frequent use of VAD in modern obstetrics, it is striking how little is known about the long-term outcomes. According to the current study, this group may be at risk for neurodevelopmental difficulties. The information available does not allow one to determine whether the VAD procedure causes these difficulties, or whether the entire delivery process contributes, since children born after an SSCD were calculated to be at a similar risk. It may be that the complete situation, including the lack of progress in the second stage of labour, and the maternal and fetal conditions, are responsible for the problematic outcome. It appears, however, that children born after VAD might have an increased risk of ADHD, as indicated by the register data. This increased incidence was not observed in the SSCD group.

### Strengths and limitations
This is a study with a well-defined cohort of children, investigating long-term outcomes using a validated screening method. By including the registered neurodevelopmental ICD-10 diagnoses at the SNBHW, the value of the study is enhanced.

Even though the validated questionnaire contains a normative reference group, we included a control group with children born via SVD to increase validity. In addition, SSCD group with a fetal head station at level 0 or

more, without a prior VAD attempt, is a clinically relevant comparison group. This group had a significantly longer duration of the second stage (a median of 5 hours) and showed similar results in the FTF as the VAD group. On the other hand, there were more children with neurodevelopmental diagnoses and NICU admissions in the VAD group.

It might be questioned whether parents' estimates of a child's development are objective. However, the FTF is not a diagnostic tool, and a multiprofessional assessment is needed to diagnose neurodevelopmental disorders in children and youth. Since the questionnaire was distributed only in the Swedish language, an inevitable selection of parents might exist, but this applies to the whole cohort.

The maternal educational level was included as a proxy for socioeconomic status, since most questionnaires were filled out by mothers. Mothers with higher educational levels more often had children scoring <90th percentile, and mothers in the VAD group had significantly lower educational level than the other two groups. This variable was adjusted for in the multivariable regression analysis.

The low response rate in the SVD group is a limitation and could be attributed to the fact that a control group is less likely to respond. This could potentially lead to selection bias, resulting in parents of children with difficulties being more likely to respond. Alternatively, it may be that parents who experienced a traumatic delivery were less prone to respond. The response analysis in the VAD group did not reveal any differences, except a higher rate of mid-station VADs in the non-respondent group, which is believed to not affect the result, except a possible risk for underestimation of FTF scores.

Including the failed VADs in this group may bias the result by including children with worse outcomes but shows the relevance in the clinical reality. On the other hand, the SSCD group was clinically more uniform for comparison. Since the SSCD group is smaller, differences may be hidden, and conclusive insight is hard to draw regarding registered diagnosis due to sparse data. To remedy the small sample size in the SSCD group, the age range was extended while still suitable for the FTF. We believe this was necessary to try to answer the research question without introducing risks.

### Interpretation
The study is observationally designed to screen for neurodevelopmental difficulties rather than to reveal causal relationships. With the use of FTF several neurodevelopmental aspects were possible to study. The results are in contrast with the few existing studies reporting on specific long-term neurodevelopmental aspects,[19–21 23 33] but these have not stratified according to the VAD type, they have smaller cohorts, and did not use validated follow-up methods.

Even though children had motor skill problems according to the FTF, they were not registered with individual motor skill diagnoses. This may rather be revealed

in a more complex form that is, as a comorbidity in ADHD. Whether or not low-station and mid-station VAD or SSCD affect specific parts of the fetal brain resulting in impaired neurodevelopment for different skills, is unknown. Although the aetiology of diagnoses such as ADHD is heterogeneous and unclear, it is known to be associated with traumatic brain injury.[34] Apart from this, there is an established association between ADHD and heredity.[35 36] Registered neurodevelopmental diagnoses among parents or siblings were not collected.

## CONCLUSION

Obstructed labour at the second stage in term deliveries is a delicate mission for clinicians. Although interpreted with caution, the significant difference in long-term neurodevelopmental difficulties in children delivered with VAD or SSCD compared with an SVD is worrying and indicates the need for increased knowledge in the field. If additional studies show similar results, it seems necessary to revise the existing guidelines for VAD and SSCD.

**Author affiliations**
¹Division of Obstetrics and Gynaecology, Karolinska Institute Department of Clinical Science Intervention and Technology, Huddinge, Sweden
²Pregnancy Care and Delivery, Karolinska University Hospital, Stockholm, Sweden
³Neuropaediatrics, Karolinska University Hospital, Stockholm, Sweden
⁴Division of Paediatrics, Karolinska Institute Department of Clinical Science Intervention and Technology, Huddinge, Sweden
⁵Karolinska Institute, Stockholm, Sweden

**Acknowledgements** The paper is part of Dr. Romero's doctoral thesis.

**Contributors** GA and MW designed the study, acquired, analysed, interpreted the data, and wrote and revised the manuscript. SR acquired, analysed, interpreted the data and wrote and revised the manuscript. KL analysed, interpreted the data and revised the manuscript. The corresponding author attests that all listed authors meet authorship criteria and that no others meeting the criteria have been omitted. GA acts as guarantor.

**Funding** This work was supported by the Centre for Innovative Medicine (grant number FoUI-954462), Karolinska Institutet, the General Maternity Hospital Foundation (grant number N/A) and Kommunfullmäktige, Stockholm Stad, grant number FoUI-953029 and FoUI-955070).

**Competing interests** None declared.

**Patient and public involvement** Patients and/or the public were not involved in the design, or conduct, or reporting, or dissemination plans of this research.

**Patient consent for publication** Not applicable.

**Ethics approval** This study involves human participants and ethical approval was obtained by the Regional Ethical Review Board Stockholm, Sweden (DNR 2014/1860-31, DNR 2017/1411-32, DNR 2019/04880 and DNR 2021/02457). Participants gave informed consent to participate in the study before taking part.

**Provenance and peer review** Not commissioned; externally peer reviewed.

**Data availability statement** Data are available on reasonable request. The corresponding author has the right to grant on behalf of all authors and does grant on behalf of all authors, if the manuscript is accepted, a worldwide licence to the publishers and its licence in perpetuity, in all forms, formats and media (whether known now or created in the future), to (1) publish, reproduce, distribute, display and store the contribution, (2) translate the contribution into other languages, create adaptations, reprints, include within collections and create summaries, extracts and/or, abstracts of the contribution, (3) create any other derivative work(s) based on the contribution, (4) to exploit all subsidiary rights in the contribution, (5) the inclusion of electronic links from the contribution to third party material wherever it may be located and (6) licence any third party to do any or all of the above.

**ORCID iD**
Stefhanie Romero http://orcid.org/0000-0001-9673-177X

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
