## [Reviewer comments · BMJ Paediatrics Open]

ARTICLE DETAILS

TITLE (PROVISIONAL)	Long-term neurodevelopmental outcome in children born after vacuum-assisted delivery compared to second stage caesarean delivery and spontaneous vaginal delivery - a cohort study
AUTHORS	Romero, Stefhanie Lindström, Katarina Listermar, Johanna Westgren, Magnus Ajne, Gunilla

VERSION 1 - REVIEW

REVIEWER	Dr. Conrad Kabali
REVIEW RETURNED	31-May-2023

GENERAL COMMENTS	Title: The title should accurately reflect the objective of the study as a comparative study. For instance, a revised title may read as follows: "Comparison of long-term neurodevelopmental outcomes in children born after vacuum-assisted delivery, caesarean delivery, and spontaneous vaginal delivery: a cohort study." Page 3, line 38: To clarify that this is a comparative study, I recommend to begin the second sentence with "In the VAD group." Page 3, line 43: Kindly replace "cohort" with "number" in this sentence. Page 3, line 45: To ensure clarity regarding the study design involving multiple groups, consider rephrasing the sentence to: "The SVD group consisted of women..." Page 3, line 53: Could you please provide further clarification? Did you mean to state that due to a small sample size, the age range was expanded while still remaining relevant to answering the research question? Please elaborate on the suitability of the modified range for screening purposes. Page 5, line 15: I advise to rephrase the statement as follows: "...and reported 95% confidence intervals and p-values." This re-wording will help address the concerns I mentioned in subsequent sections of your manuscript. Page 5, line 24: Please replace the term "performed" with either "used" or "employed" to enhance clarity. Page 5, line 38: Are there any concerns regarding selection bias due to a low response rate? Please discuss this potential issue. Table 3: To reiterate a previous comment, it is important not to interpret the
---

	results solely based on "statistical significance." Merely considering an odds ratio (OR) of 1.7 as "significant" while disregarding an OR of 2 or 0.5 is not meaningful. The large p-values may be a consequence of inadequate sample size rather than evidence of "no difference," as further supported by the confidence intervals. Please be cautious in the interpretation of the results. Page 9, line 28: This example further underscores the potential issues that may arise when relying solely on "statistical significance" to interpret the results. While comparing 8% vs. 2% in both cases, your interpretation suggests a difference between VAD (8%) and SVD (2%), but no difference between VAD (8%) and ECD (2%). It is worth noting that there are only 2 observations in the ECD group, so a larger p-value in this instance reflects imprecision rather than the absence of a difference. Additionally, please report the corresponding confidence intervals. Table 5: Regarding Table 5, it is essential to address the concern regarding the feasibility of running the logistic model with 0 to 5 observations in certain arms. Particularly, considering the adjustment for several confounders, it is crucial to ensure that the analysis performed is valid for the sample size. Kindly revisit your model and verify that the analysis aligns appropriately with the available data.
--	--

REVIEWER	Dr. Giulia Muraca McMaster University, Obstetrics and Gynecology and Health Research Methods, Evidence and Impact
REVIEW RETURNED	01-Jun-2023

GENERAL COMMENTS	This cohort study aimed to compare neurodevelopment in children born to mothers with a vacuum delivery, spontaneous vaginal delivery (SVD) and emergency cesarean delivery (CD) in children born in two delivery wards in a university hospital in Sweden. The main outcome was a validated neurodevelopmental screening tool, the Five-to-Fifteen (FTF) questionnaire. Crude and adjusted regression analyses were used to estimate the associations between mode of delivery in the second stage of labour (vacuum, spontaneous vaginal, or CD) and neurodevelopmental difficulties. The crude results showed children born after vacuum delivery and second stage CD had an increased rate of neurodevelopmental difficulties in motor skills and perception compared to SVD. Multivariable regression analysis showed an increased odds of problems with motor skills when delivered with vacuum or second stage CD compared with SVD. This is an important topic and the authors are correct in their assertion that there is a paucity of information on these relationships. The manuscript has potential yet lacks clarity and there are some methodologic peculiarities that need to be addressed. Specific comments to help achieve this end are included below by study section. Abstract The abstract needs refinement. The objective should be more succinct and the description of patients and the outcomes measures lack clarity. For example, it is stated that the participants included 1455 children, however the primary outcome was measured among
--

586 children. Further, regression analyses do not compare results, regression analyses are employed to estimate associations between your independent and dependent variable.

There are inconsistencies in the abstract and the manuscript text. The abstract states that the study included 440 children born by vacuum (years 2007-2009), 844 children born after a SVD (years 2010-2012), and 171 children born via an ECD (years 2007-2014). The Methods section state a final cohort (after exclusions) of 468 children born by vacuum, 869 born by SVD and 184 by 2nd stage CD. Lastly, the Results section and Tables report outcomes on 253 VDs, 247 SVDs and 86 2nd stage CDs. Please clarify these discrepancies and make this easy for the reader to understand.

Introduction

Consider renaming your CD group “second stage CD” or “CD at full dilatation” and move away from the ECD acronym. ECD gives the impression that you have included all emergent CDs when, in fact, you have correctly restricted this group to only second stage CD. Calling these ECDs in the second stage is redundant since it is clearly an emergent CD if the patient has had a trial of labour that has progressed to second stage.

Page 2, Line 8: The relevant measure of frequency for vacuum delivery is incidence (not prevalence).

Page 2, Line 9: The proportion of vacuum deliveries you are describing (3% in the US and 5-10% in Europe, Canada, Australia/NZ) represents the percentage of vacuum delivery among all vaginal deliveries, not among operative vaginal deliveries.

Page 2, Line 11: Beyond stating that they are rare, include the incidence of severe birth trauma among vacuum-assisted deliveries.

Page 2, Line 14: Clarify what is meant by ‘other instrumental delivery modes’. Forceps? obstetric spatula? Sequential application of multiple instruments? Additionally, this statement regarding inconsistent finding needs citations.

Line 15: If you wish to cite previous studies that have compared vacuum delivery with all emergency CD (a group contaminated by deliveries that are not alternatives to vacuum and thus inappropriate), include a statement to acknowledge this limitation alongside any relative outcome estimates.

The final sentence (lines 31-32) should be removed from the introduction and shifted to the Methods section.

Methods

It is not clear whether the data were collected prospectively or retrospectively or cross-sectionally. Line 37 suggests this is a prospective cohort design, yet line 50 (pg 2) alludes to data access and chart review suggesting a retrospective cohort design. Finally, the results of the FTF questionnaire are cross-sectional.

A study schematic showing the ascertainment window, follow up period, and administration of the FTF for each arm of the study would help to clarify the study design.

Please include a flowchart that shows the derivation of the cohort and exclusions. The manuscript text alludes to a Figure S1 but I cannot see it.

Please include a figure (perhaps as supplementary material) illustrating your DAG.

Provide your rationale for using two reference groups (SVDs and 2nd stage CDs)? I can infer why you have done with but it should be explicit and not misinterpreted.

Explain to the reader why the children delivered via SVDs were included from a different time period than the vacuum deliveries?

Line 47: the SVD group was delivered 'the same date and time' as the vacuum delivery group, but three years later. Does this mean that these deliveries were matched by calendar date and time of birth? The fact that these SVDs occurred three years later seems to contradict this earlier sentence.

How do you suppose including child's age as a covariate may have affected your adjusted estimates? For example, in the comparison between vacuum deliver and second stage CD, what do you think would happen to those second stage CD outside of the 10-13 year age range? Would they contribute to the relative estimate of association? Do you think your findings can be generalized to children outside the overlapping age range?

Lines 5-9 on pg 3 should be shifted into the Results section.

Please clarify which version of the FTF was used, the one developed for use in children between five and 17 years of age?

Please specify the type of multivariable regression analysis. You report odds ratios, so I am assuming it is a logistic regression analysis but this should be included in your Analysis section.

Results

The response rate in the SVD group was lower than in the vacuum and CD groups. It is thus unclear why the response analysis in Table 1 focused on vacuum delivery (or if similar response analyses for SVD and CD were not included). This analysis was not described in the Methods – was this conceived post-hoc?

Present the distribution of children's age in each group, preferably by year (or in the case of the CD group by 2-year categories) and not using mean/median descriptive statistics.

It is unconventional to present the crude relative estimates in tabular format and include the adjusted estimates in the text. It would be presented more clearly if you included the adjusted estimates in Table 3 in the last column (and remove the p-value column, which are redundant since the confidence intervals are reported).

Discussion

Page 9, Line 42: Including a variable indicating the recorded indication for operative delivery does not address confounding by indication (see Joseph, K.S., Mehrabadi, A. & Lisonkova, S.

	Confounding by Indication and Related Concepts. Curr Epidemiol Rep, 2014;1:1–8.) Page 10, line 33: If the aim of the study is not to reveal causal relationships, the use of a DAG is puzzling. If an 2nd stage CD was performed after an attempted vacuum delivery, was this included in the CD group or the vacuum group? If the former (included in the CD group), or if these failed attempts at vacuum deliveries were excluded from your cohort, how might this bias your results? How might the shift in time for the SVD reference group and the inclusion of deliveries from a second centre for the 2nd stage CD reference group affect your results? Have you considered that your ability to detect a difference in the rate of children scoring ≥ 90th percentile in the FTF may be hindered by a type II error? Several limitations to this analysis (described above) should be acknowledged in the limitation section.
--	--

VERSION 1 – AUTHOR RESPONSE

Reviewer: 1

Dr. Conrad Kabali

- The title should accurately reflect the objective of the study as a comparative study. For instance, a revised title may read as follows: " Comparison of long-term neurodevelopmental outcomes in children born after vacuum-assisted delivery, caesarean delivery, and spontaneous vaginal delivery: a cohort study."

oWe are grateful for this valuable suggestion. We agree and have suggested a more informative title for the readers: "Long-term neurodevelopmental outcome in children born after vacuum-assisted delivery compared to second stage caesarean delivery and spontaneous vaginal delivery: a cohort study". The primary focus and aim were to evaluate if there are any signs of long-term effects after low and mid station vacuum assisted delivery. However, when the design of the study was done, we realized that normal vaginal delivery would not be the proper comparison, rather emergency caesarean section at second stage. The title however was unfortunately not changed from the first draft of study protocol. To manage to define proper individuals delivered by emergency caesarean section at second stage, the study-design had to be a cross sectional cohort in a single centre delivery ward, as each patient file had to be scrutinized for accurate inclusion.

- Page 3, line 38: To clarify that this is a comparative study, I recommend to begin the second sentence with "In the VAD group."

oThank you for your comment. Please see explanation above.

- Page 3, line 43: Kindly replace "cohort" with "number" in this sentence.

oWe have replaced "cohort" with "number"

- Page 3, line 45: To ensure clarity regarding the study design involving multiple groups, consider rephrasing the sentence to: "The SVD group consisted of women..."

oWe have changed this accordingly.

•Page 3, line 53: Could you please provide further clarification? Did you mean to state that due to a small sample size, the age range was expanded while still remaining relevant to answering the research question? Please elaborate on the suitability of the modified range for screening purposes.

oWe are grateful for the comment in need of clarification. Please see the added text on page 3, lines 50-54, and page 12, lines 20-26.

•Page 5, line 15: I advise to rephrase the statement as follows: "...and reported 95% confidence intervals and p-values." This re-wording will help address the concerns I mentioned in subsequent sections of your manuscript.

oThank you, it has been rephrased accordingly.

•Page 5, line 24: Please replace the term "performed" with either "used" or "employed" to enhance clarity.

oThe term "performed" was changed to "used".

•Page 5, line 38: Are there any concerns regarding selection bias due to a low response rate? Please discuss this potential issue.

oPlease see a discussion about this on page 12, lines 10-16.

•Table 3: To reiterate a previous comment, it is important not to interpret the results solely based on "statistical significance." Merely considering an odds ratio (OR) of 1.7 as "significant" while disregarding an OR of 2 or 0.5 is not meaningful. The large p-values may be a consequence of inadequate sample size rather than evidence of "no difference," as further supported by the confidence intervals. Please be cautious in the interpretation of the results.

oWe are grateful for the clarification and agree regarding the interpretation of the results. We have rephrased this sentence, please see page 11 lines 28-35.

•Page 9, line 28: This example further underscores the potential issues that may arise when relying solely on "statistical significance" to interpret the results. While comparing 8% vs. 2% in both cases, your interpretation suggests a difference between VAD (8%) and SVD (2%), but no difference between VAD (8%) and ECD (2%). It is worth noting that there are only 2 observations in the ECD group, so a larger p-value in this instance reflects imprecision rather than the absence of a difference. Additionally, please report the corresponding confidence intervals.

oYes, we agree and have rephrased this section on page 9, lines 49-52, and in the limitation section on page 12, line 19-26.

oWe have now added OR and the 95%CI in table 5.

•Table 5: Regarding Table 5, it is essential to address the concern regarding the feasibility of running the logistic model with 0 to 5 observations in certain arms. Particularly, considering the adjustment for several confounders, it is crucial to ensure that the analysis performed is valid for the sample size. Kindly revisit your model and verify that the analysis aligns appropriately with the available data.

oTable 5 shows the secondary outcomes regarding ICD-10 diagnosis for children with objectively neurodevelopmental disability according to the Swedish health register and diagnosis done by health care professionals (in this case subspecialised paediatricians in Sweden). A logistic model is not undertaken for this data due to the concerns also addressed by the reviewer, but we now present an unadjusted OR with a 95%CI as a descriptive analysis instead of p-values.

Reviewer: 2

This cohort study aimed to compare neurodevelopment in children born to mothers with a vacuum delivery, spontaneous vaginal delivery (SVD) and emergency cesarean delivery (CD) in children born in two delivery wards in a university hospital in Sweden. The main outcome was a validated neurodevelopmental screening tool, the Five-to-Fifteen (FTF) questionnaire. Crude and adjusted regression analyses were used to estimate the associations between mode of delivery in the second stage of labour (vacuum, spontaneous vaginal, or CD) and neurodevelopmental difficulties. The crude results showed children born after vacuum delivery and second stage CD had an increased rate of neurodevelopmental difficulties in motor skills and perception compared to SVD. Multivariable regression analysis showed an increased odds of problems with motor skills when delivered with vacuum or second stage CD compared with SVD.

This is an important topic and the authors are correct in their assertion that there is a paucity of information on these relationships. The manuscript has potential yet lacks clarity and there are some methodologic peculiarities that need to be addressed. Specific comments to help achieve this end are included below by study section.

We are grateful for the valuable comments and the assessment that there is a need for increased knowledge in the field.

We have tried to respond and improve the manuscript according to the answers below.

Abstract

- The abstract needs refinement. The objective should be more succinct and the description of patients and the outcomes measures lack clarity. For example, it is stated that the participants included 1455 children, however the primary outcome was measured among 586 children. Further, regression analyses do not compare results, regression analyses are employed to estimate associations between your independent and dependent variable.

- o Thank you for your comments. Please see changes in the abstract on lines 2-4, 11-14, and 19-21.

- There are inconsistencies in the abstract and the manuscript text. The abstract states that the study included 440 children born by vacuum (years 2007-2009), 844 children born after a SVD (years 2010-2012), and 171 children born via an ECD (years 2007-2014). The Methods section state a final cohort (after exclusions) of 468 children born by vacuum, 869 born by SVD and 184 by 2nd stage CD. Lastly, the Results section and Tables report outcomes on 253 VDs, 247 SVDs and 86 2nd stage CDs. Please clarify these discrepancies and make this easy for the reader to understand.

- o We have tried to clarify this further. Please see our study flowchart illustrated in Figure 1.

Introduction

- Consider renaming your CD group “second stage CD” or “CD at full dilatation” and move away from the ECD acronym. ECD gives the impression that you have included all emergent CDs when, in fact, you have correctly restricted this group to only second stage CD. Calling these ECDs in the second stage is redundant since it is clearly an emergent CD if the patient has had a trial of labour that has progressed to second stage.

- o Thank you for your suggestion. We have now decided to name it “second stage caesarean delivery (SSCD)”.

- Page 2, Line 8: The relevant measure of frequency for vacuum delivery is incidence (not prevalence).
 - o This is correct and we have changed the term to incidence.
- Page 2, Line 9: The proportion of vacuum deliveries you are describing (3% in the US and 5-10% in Europe, Canada, Australia/NZ) represents the percentage of vacuum delivery among all vaginal deliveries, not among operative vaginal deliveries.
 - o Please see the changes on page 3, line 9.
- Page 2, Line 11: Beyond stating that they are rare, include the incidence of severe birth trauma among vacuum-assisted deliveries.
 - o Please see the information added on page 3, lines 12-13.
- Page 2, Line 14: Clarify what is meant by 'other instrumental delivery modes'. Forceps? obstetric spatula? Sequential application of multiple instruments? Additionally, this statement regarding inconsistent finding needs citations.
 - o We refer to forceps and obstetric spatula. We have included citations, please see page 3 line 15.
- Line 15: If you wish to cite previous studies that have compared vacuum delivery with all emergency CD (a group contaminated by deliveries that are not alternatives to vacuum and thus inappropriate), include a statement to acknowledge this limitation alongside any relative outcome estimates.
 - o We have only included citations where the studies include CD at second stage, please see page 3 lines 15-21.
- The final sentence (lines 31-32) should be removed from the introduction and shifted to the Methods section.
 - o Thank you, these lines have been removed.

Methods

- It is not clear whether the data were collected prospectively or retrospectively or cross-sectionally. Line 37 suggests this is a prospective cohort design, yet line 50 (pg 2) alludes to data access and chart review suggesting a retrospective cohort design. Finally, the results of the FTF questionnaire are cross-sectional.
 - o We agree and have changed this to cross-sectional.
- A study schematic showing the ascertainment window, follow up period, and administration of the FTF for each arm of the study would help to clarify the study design.
 - o Please see added Figure 1: study flowchart
- Please include a flowchart that shows the derivation of the cohort and exclusions. The manuscript text alludes to a Figure S1 but I cannot see it.
 - o Please see the new included Figure 1: study flowchart.
- Please include a figure (perhaps as supplementary material) illustrating your DAG.

- o Please see attached supplementary figure S1
- Provide your rationale for using two reference groups (SVDs and 2nd stage CDs)? I can infer why you have done with but it should be explicit and not misinterpreted.
- o We have tried to clarify this on page 3, lines 18-21.
- Explain to the reader why the children delivered via SVDs were included from a different time period than the vacuum deliveries?
- o This study was initially meant to look at children delivered with a low or mid station VAD as the primary group. When the preliminary results showed a difference in the FTF between children delivered with a VAD and the comparative group (included in the FTF), we realized that we had to expand the comparative groups to increase validity (that is a clinically relevant group (SSCD) and a normative group (SVD) from the same delivery unit). Please see page 11, lines 46-51.
- Line 47: the SVD group was delivered 'the same date and time' as the vacuum delivery group, but three years later. Does this mean that these deliveries were matched by calendar date and time of birth? The fact that these SVDs occurred three years later seems to contradict this earlier sentence.
- o Thank you, this has been rephrased in page 3, lines 48-49.
- How do you suppose including child's age as a covariate may have affected your adjusted estimates? For example, in the comparison between vacuum deliver and second stage CD, what do you think would happen to those second stage CD outside of the 10-13 year age range? Would they contribute to the relative estimate of association? Do you think your findings can be generalized to children outside the overlapping age range?
- o Thank you for your comments. We have now included a table (Table 2) with the age distributions of the children according to mode of delivery.
- o We have eliminated the child's age as a covariate in the multivariate logistic regression analysis since we have rethought and believe this to be a collider given that the questionnaire is already adjusted by age.
- Lines 5-9 on pg 3 should be shifted into the Results section.
- o This information has been added to the result section.
- Please clarify which version of the FTF was used, the one developed for use in children between five and 17 years of age?
- o We have retyped this sentence since we have used the FTF questionnaire, suitable for all our included ages (six to 13 years of age).
- Please specify the type of multivariable regression analysis. You report odds ratios, so I am assuming it is a logistic regression analysis but this should be included in your Analysis section.
- o Thank you, the term logistic has been added to specify the type of multivariate regression analysis

Results

- The response rate in the SVD group was lower than in the vacuum and CD groups. It is thus unclear why the response analysis in Table 1 focused on vacuum delivery (or if similar response analyses for SVD and CD were not included). This analysis was not described in the Methods – was this conceived post-hoc?
 - o As mentioned above, this study was initially meant to look at children delivered with a low or mid station VAD as the primary group. When the preliminary results showed a difference in the FTF between children delivered with a VAD and the comparative group (included in the FTF), we realized that we had to expand the comparative groups to increase validity. The response analysis was already predefined in the study protocol.
 - o Please see page 5, line 6.
 - o To further clarify this, table 1 has been changed to a supplementary table, S1.
- Present the distribution of children's age in each group, preferably by year (or in the case of the CD group by 2-year categories) and not using mean/median descriptive statistics.
 - o Thank you, we agree. Please see the newly added Table 2.
- It is unconventional to present the crude relative estimates in tabular format and include the adjusted estimates in the text. It would be presented more clearly if you included the adjusted estimates in Table 3 in the last column (and remove the p-value column, which are redundant since the confidence intervals are reported).
 - o Thank you for your suggestion. We have taken this into consideration and made the appropriate changes to improve table 3.

Discussion

- Page 9, Line 42: Including a variable indicating the recorded indication for operative delivery does not address confounding by indication (see Joseph, K.S., Mehrabadi, A. & Lisonkova, S. Confounding by Indication and Related Concepts. *Curr Epidemiol Rep*, 2014;1:1–8.)
 - o Thank you for this clarification and for the suggested article. We have decided to delete this sentence.
- Page 10, line 33: If the aim of the study is not to reveal causal relationships, the use of a DAG is puzzling.
 - o Using this study design, where we look at long-term outcomes, we find it difficult to reveal any causal relationships, but tried to approximate this as much as possible. The use of a multivariate logistic regression analysis is one way and to do this, the use of a DAG is helpful.
- If an 2nd stage CD was performed after an attempted vacuum delivery, was this included in the CD group or the vacuum group? If the former (included in the CD group), or if these failed attempts at vacuum deliveries were excluded from your cohort, how might this bias your results?
 - o 13% of all VADs failed. These were later delivered by forceps (n =8), a forceps attempt, and then SSCD (n =8) or by SSCD directly (n =17). Since they were all subjected to the vacuum extractor, and the focus of this study initially was to investigate the outcome after a low- or mid-station VAD, they were all included in the VAD group. This could of course bias our results by having children with worse outcomes in the VAD group. But since the SSCD group is overall smaller, differences may be hidden anyway. We believe the main result of this study is that a traumatic delivery, either by VAD or

SSCD, causes long-term outcomes in these children. We have tried to clarify this, please see page 8, lines 3-6.

- How might the shift in time for the SVD reference group and the inclusion of deliveries from a second centre for the 2nd stage CD reference group affect your results?

- o During the overall study period (2007-2014) we did not see any changes in work performance of structure at these two centra, both which are included in the same organization within Karolinska University Hospital. Because of this, we do not believe the shift in time had a significant impact on the results.

- Have you considered that your ability to detect a difference in the rate of children scoring ≥ 90 th percentile in the FTF may be hindered by a type II error?

- o Thank you for highlighting this issue, and we absolutely agree, mainly due to the sample sizes, the possible response-bias, and the complexity of the situation where the VAD group also contains children delivered by SSCD or forceps. We have tried to clarify this further in the revised manuscript, please see page 13, lines 21-23.

- Several limitations to this analysis (described above) should be acknowledged in the limitation section.

- o Thank you for all your suggestions, please see the revised parts in the discussion section, page 11 lines 9-26

VERSION 2 – REVIEW

REVIEWER	Dr. Conrad Kabali
REVIEW RETURNED	15-Aug-2023

GENERAL COMMENTS	Page 5, line 11: Kindly substitute "multivariate" with "multivariable" throughout the manuscript. Page 8, Table 5: Evidently, there exists an issue of sparse data, and the findings derived from a logistic model may not be deemed reliable, except possibly within the domain of "neurodevelopmental diagnosis." I advise to mention that conclusive insights cannot be drawn from Table 5 owing to insufficient data, and consequently, you should omit the last two columns.
---

VERSION 2 – AUTHOR RESPONSE

Page 5, line 11: Kindly substitute "multivariate" with "multivariable" throughout the manuscript. Thank you for this correction. The changes have been made throughout the manuscript.

Page 8, Table 5: Evidently, there exists an issue of sparse data, and the findings derived from a logistic model may not be deemed reliable, except possibly within the domain of "neurodevelopmental diagnosis." I advise to mention that conclusive insights cannot be drawn from Table 5 owing to insufficient data, and consequently, you should omit the last two columns. Thank you, and as suggested, we have now omitted the last two columns in table 5 and added a comment regarding the sparse data under the section "strength and limitations" on page 9 (please see the revised manuscript marked copy).